# Synthesis and Biological Evaluation of BODIPY-PF-543

**DOI:** 10.3390/molecules24234408

**Published:** 2019-12-02

**Authors:** Jitendra Shrestha, Gil Tae Hwang, Taeho Lee, Seon Woong Kim, Yoon Sin Oh, Yongseok Kwon, Seung Woo Hong, Sanghee Kim, Hong Seop Moon, Dong Jae Baek, Eun-Young Park

**Affiliations:** 1College of Pharmacy, Mokpo National University, Jeonnam 58554, Korea; shresthasimon2011@mokpo.ac.kr (J.S.); tjsdnd123@mokpo.ac.kr (S.W.K.); hbsmoon@mokpo.ac.kr (H.S.M.); 2Department of Chemistry, Kyungpook National University, Daegu 41566, Korea; giltae@knu.ac.kr (G.T.H.); tmddnzkf1@naver.com (S.W.H.); 3College of Pharmacy, Research Institute of Pharmaceutical Sciences, Kyungpook National University, Daegu 41566, Korea; tlee@knu.ac.kr; 4Department of Food and Nutrition, Eulji University, Seongnam 13135, Korea; ysoh@eulji.ac.kr; 5Department of Chemistry, Sogang University, Seoul 04107, Korea; ykwon@sogang.ac.kr; 6College of Pharmacy, Seoul National University, Seoul 08826, Korea; pennkim@snu.ac.kr

**Keywords:** sphingosine kinase, PF-543, BODIPY, inhibitor, confocal microscopy

## Abstract

Sphingosine-1-phosphate (S1P) regulates the proliferation of various cells and promotes the growth of cancer cells. Sphingosine kinase (SK), which transforms sphingosine into S1P, has two isotypes: SK1 and SK2. To date, both isotypes are known to be involved in the proliferation of cancer cells. PF-543, an SK1 inhibitor developed by Pfizer, strongly inhibits SK1. However, despite its strong SK1 inhibitory effect, PF-543 shows low anticancer activity in vitro. Therefore, additional biological evidence on the anticancer activity of SK1 inhibitor is required. The present study aimed to investigate the intracellular localization of PF-543 and identify its association with anticancer activity by introducing a fluoroprobe into PF-543. Boron–dipyrromethene (BODIPY)-introduced PF-543 has a similar SK1 inhibitory effect as PF-543. These results indicate that the introduction of BODIPY does not significantly affect the inhibitory effect of SK1. In confocal microscopy after BODIPY-PF-543 treatment, the compound was mainly located in the cytosol of the cells. This study demonstrated the possibility of introducing fluorescent material into an SK inhibitor and designing a synthesized compound that is permeable to cells while maintaining the SK inhibitory effect.

## 1. Introduction

Sphingolipids, which have been studied for years in various fields (e.g., biology and biochemistry), act as signals between cells; in fact, even a small amount of sphingolipids can regulate various biological reactions. They are considered the most structurally diverse class of lipids, and their mechanism of action in vivo is markedly complex. Sphingolipids are transformed into various forms by enzymes, and they play a unique role. Ceramides also have a variety of structures depending on the type of fatty acid, and their functions are complicated. Sphingosine is transformed into sphingosine-1-phosphate (S1P) by sphingosine kinase (SK) and cell growth is subsequently induced [1]. Several studies have demonstrated the association between S1P and the proliferation of cancer cells, and the abnormal increase of S1P has been detected in patients with cancer [2]. SK inhibitors have been developed based on biological studies showing that the growth of cancer cells can be inhibited by regulating SKs [3], which have two isotypes—SK1 and SK2. These isotypes act at different positions within cells but both induce S1P production and are agents in the growth of cancer cells [4]. To date, selective inhibitors for SK1 and SK2 have been developed, but the relationship between their structure and selectivity remains unclear. The X-ray crystal structure of SK1 has been revealed and used for the development of various SK1 inhibitors [5]; however, the structure of SK2 remains unknown.

Nevertheless, ABC294640 (Figure 1), a selective inhibitor of SK2 developed by Apogee Biotechnology Corp., is currently undergoing phase II clinical trials involving various solid cancers [6]. SK1 inhibitors initially had a sphingosine form such as dimethyl sphingosine (DMS), but it was difficult to use them as drugs because of their low solubility in water owing to their long aliphatic chain structure. Therefore, SK inhibitors which are structurally similar to FTY720 have been developed [7]. FTY720 (Gilenya, Figure 1), a treatment for multiple sclerosis (MS) developed by Novartis, is a functional agonist or antagonist for S1P receptor (S1PR) and it may inhibit SK1 [8]. FTY720 also inhibits cancer cells by activating protein phosphatase 2A (PP2A) [9]. The antitumor effect of FTY720 in cancer cells including breast, liver and uterine cancer cells is based on S1PR-independent pathway [10]. The phosphorylated FTY720 (FTY720-P) is unable to induce cell death, proving that activation of S1PR is not sufficient for cell death [11]. Although the anticancer property of FTY720 is not clinically proven and therefore not used as an anticancer agent, FTY720 successfully sold in the pharmaceutical market for MS therapy, and is one of the best-selling products manufactured by Novartis. This success has engaged several pharmaceutical companies in the development of drugs using sphingolipid regulators, and many substances related to S1P modulators are currently undergoing clinical trials [12]. SK inhibitors can be divided into two large groups: Lipid and nonlipid structures [3]. The most potent inhibitor of SK1 in nonlipid structures is PF-543 developed by Pfizer Inc. (Figure 1). PF-543 inhibits SK1 with a half-maximal inhibitory concentration (IC_50_) of 2.0 nM [13]. Further research by Pfizer revealed that the unique benzenesulfonyl group of PF-543 is an essential structure for SK1 inhibition [14]. This result has significantly contributed to improving the existing lipid structures of SK inhibitors, which cannot be easily developed into drugs. However, although PF-543 effectively reduces S1P, its inhibitory effects on cancer cells are unsatisfactory [15]. This is inferred to be related to the fact that few animal studies have been reported despite the fact that PF-543 has a strong inhibitory effect on SK1. These issues raise questions about the development of anticancer drugs using SK1 inhibitors. Therefore, several future biological studies are required to clarify the relationship between SK1 inhibitors and the growth of cancer cells. To do this, we introduced a fluoroprobe into PF-543 and demonstrated that this molecule is applicable to biological research. We have synthesized a compound in which dansyl was introduced into PF-543 (Dansyl-PF-543, Figure 1) and reported the results. Although dansyl-PF-543 showed a maximum fluorescence value of 470 nm and an SK1 inhibition effect of IC_50_ = 12.3 nM, no clear evaluations were conducted in further studies using fluorescence microscopy [16]. It is necessary to introduce another probe to observe intracellular distribution using fluorescence microscopy. BODIPY is one of the excellent probes with high absorption and fluorescence in the visible range, photochemical stability, and chemical robustness [17,18]. BODIPY-based probes have low toxicity and are stable at physiological pH, thus making them excellent probes for use in living cells [19,20]. Therefore, we introduced BODIPY, which generally shows a fluorescence value of 500 nm, into PF-543. Extensive docking studies of SK1 in several laboratories have demonstrated that a structurally bulky group can be docked at the tail of SK1 inhibitors [21]. Therefore, we hypothesized that the bulky structure of BODIPY does not significantly affect its SK inhibitory effect. In addition, BODIPY is a fluoroprobe that has been introduced into lipids and sphingolipids for years, and studies have demonstrated its high membrane permeability.

## 2. Results and Discussion

### 2.1. Chemical Synthesis

We first used a known Compound **3** [13] as a starting material to synthesize BODIPY-PF-543 (Scheme 1). We added sodium azide to this compound to synthesize Compound **4** and introduced azide into Compound **4**. Subsequently, it was reacted with 4-(bromomethyl)benzaldehyde using potassium carbonate as the base to synthesize Compound **5** into which aldehyde was introduced. Then, we synthesized Compound **6** by using it via a reductive amination. The synthesis of Compound **6** as mentioned above has already been previously reported through the synthesis of dansyl-PF-543 [16]. We reacted Compounds **6** and **7**, which was synthesized using a known synthesis method [22] via a click reaction to finally synthesize Compound **2** (BODIPY-PF-543) comprising triazole.

### 2.2. Absorption and Emission Properties of BODIPY-PF-543

We measured the UV absorbance (Cary 100 UV–vis spectrophotometer; Agilent Technologies, Santa Clara, CA, USA) and fluorescence of the synthesized BODIPY-PF-543 (Figure 2). Figure 2 shows the absorption and fluorescence spectra of Compound **2** in methanol. The absorption and emission maxima of Compound **2** appeared at 499 and 508 nm, respectively. The fluorescence quantum yields (*Φ*_F_) of Compound **2** using a 0.1 M aqueous NaOH solution of fluorescein (*λ*_ex_ = 492 nm) as the standard [23] was calculated as 0.38.

### 2.3. SK Activity Assay of PF-543 and BODIPY-PF-543

Whether the synthesized Compound **2** inhibited SK1 was analyzed in comparison with PF-543 (Figure 3). The IC_50_ values of BODIPY-PF-543 and PF-543 were 19.92 and 11.24 nM, respectively, indicating that the introduction of BODIPY into PF-543 does not significantly affect SK1 inhibition. Moreover, the effect of the BODIPY-PF-543 and PF-543 on SK2 activity was reduced by 9% and 11% at 40 μM, respectively, indicating that they selectively act on SK1 rather than SK2 (Appendix A). We examined cell viability in A549 cells, and the two compounds were observed to have cytotoxic effects (Appendix A).

### 2.4. Confocal Microscopy of BODIPY-PF-543

We observed the intracellular localization of Compound **2** using a confocal microscope (Figure 4). We used a compound concentration of 10 μM and conducted observational analysis after fixing the cells 30 min after compound treatment. As a result, Compound **2** was distributed in the cytosolic fraction in the cells. This result coincides with the location in which the known SK1 acts primarily [24].

## 3. Materials and Methods

### 3.1. Synthesis in General

Reagents used in the reaction were used by purchasing a commercially available reagent. The reaction was carried out in nitrogen state and the progress of the reaction was confirmed by TLC (silica gel 60 F254). Column chromatography was performed on silica gel grade 60 (230–400 mesh). All solvents used in the reaction were commercially available anhydrous solvents. ^1^H-NMR and ^13^C-NMR used JEOL ECZ500R (JEOL Co., Tokyo, Japan) and were measured using deuterated solvents at 500 and 125 MHz, respectively. High resolution mass spectra were measured using an Agilent Technologies G6520A Q-TOF mass spectrometer instrument using electrospray ionization (ESI).

### 3.2. Synthesis

*3-(Azidomethyl)-5-methylphenol* (**4**), Compound **3** (1.9 g, 0.0078 mol) was dissolved in DMF (40 mL), NaN_3_ (1.52 g, 0.23 mol) was added, and the mixture was stirred at 60 °C for 12 h. Water was added to stop the reaction, and it was concentrated under reduced pressure after EtOAc extraction and MgSO_4_ drying. The mixture was separated by column chromatography (*n*-hexane:EtOAc = 20:1) to give Compound **4** (895 mg, 70%): ^1^H-NMR (500 MHz, CDCl_3_) δ 6.69 (s, 1H), 6.62 (s, 1H), 6.59 (s, 1H), 4.22 (s, 2H), 2.29 (s, 3H); ^13^C-NMR (125 MHz, CDCl_3_) δ 155.9, 140.6, 136.9, 121.5, 116.2, 112.3, 54.7, 21.40; ESI-HRMS [M + H]^+^
*m*/*z* calcd for C_8_H_10_N_3_O 164.0824, found 164.0848.

*4-((3-(Azidomethyl)-5-methylphenoxy)methyl)benzaldehyde* (**5**), Compound **4** (200 mg, 1.23 mmol) was dissolved in THF (15 mL), K_2_CO_3_ (508 mg, 3.68 mmol) and 4-(bromomethyl)benaldehyde (293 mg, 1.47 mmol) were added thereto, and the mixture was stirred at 50 °C for 12 h. Water was added to stop the reaction, and it was concentrated under reduced pressure after EtOAc extraction and MgSO_4_ drying. The reaction was washed with *n*-hexane/EtOAc (5/1) to give compound **5** (271 mg, 78%): ^1^H-NMR (500 MHz, CDCl_3_) δ 10.01 (s), 7.90 (d, *J* = 8.2 Hz, 2H), 7.59 (d, *J* = 8.1 Hz, 2H), 6.76 (s, 1H), 6.75 (s, 1H), 6.73 (s, 1H), 5.13 (s. 2H), 4.26 (s, 2H), 2.33 (s, 3H); ^13^C-NMR (125 MHz, CDCl_3_) δ 192.1, 158.8, 144.0, 140.4, 136.9, 136.0, 130.2, 127.6, 122.1, 115.6, 111.6, 69.2, 54.8, 21.6; ESI-HRMS [M + H]^+^
*m*/*z* calcd for C_16_H_16_N_3_O_2_ 282.1243, found 282.1254.

*(R)-(1-(4-((3-(Azidomethyl)-5-methylphenoxy)methyl)benzyl)pyrrolidin-2-yl)methanol* (**6**), Compound **5** (180 mg, 0.64 mmol) was dissolved in 1,2-dicholroethane (10 mL), and (R)-(−)-prolinol (194 mg, 1.92mmol) and sodium triacetoxyborohydride (STB) (272 mg, 1.28 mmol) were added thereto. The mixture was stirred for 12 h at room temperature. The reaction was terminated with water and EtOAc, dried over MgSO_4_ and concentrated under reduced pressure. The mixture was separated by column chromatography (CH_2_Cl_2_:MeOH = 10:1) to give compound **6** (176 mg, 75%): ^1^H-NMR (500 MHz, CDCl_3_) δ 7.56 (d, *J* = 8.1 Hz, 2H), 7.44 (d, *J* = 8.1 Hz, 2H), 6.74 (s,1H), 6.72 (s, 1H), 6.69 (s, 1H), 5.03 (s, 2H), 4.36 (d, *J* = 13.1 Hz, 1H), 4.24 (s, 2H), 4.04 (d, *J* = 13.1 Hz, 1H), 3.79 (d, *J* = 4.6 Hz, 2H), 3.44 – 3.36 (m, 2H), 2.82 (dt, *J* = 11.0, 7.9 Hz, 1H), 2.31 (s, 3H), 2.08–1.82 (m, 4H); ^13^C-NMR (125 MHz, CDCl_3_) δ 159.0, 140.3, 138.5, 136.8, 131.2, 128.0, 121.9, 115.6, 111.6, 69.4, 68.0, 61.1, 58.7, 54.8, 53.9, 26.6, 23.4, 21.6; ESI-HRMS [M + H]^+^
*m*/*z* calcd for C_21_H_27_N_4_O_2_ 367.2134, found 367.2178. *(R)-(1-(4-((3-((4-(4-(5,5-Difluoro-1,3,7,9-tetramethyl-5H-4l4,5l4-dipyrrolo[1,2-c:2’,1’-f][1,3,2]diazaborinin-10-yl)phenyl)-1H-1,2,3-triazol-1-yl)methyl)-5-methylphenoxy)methyl)benzyl)pyrrolidin-2-yl)methanol* (**2**) Compound **7** (26 mg, 0.074 mmol) was dissolved in *t*-BuOH/H_2_O (1/1, 10 mL), and **6** (41 mg, 0.112 mmol), Na-ascorbate (22 mg, 0.112 mmol), and CuSO_4_ (18 mg, 0.112 mmol) were added thereto. The mixture was stirred for 12 h at room temperature. The reaction was terminated with water and EtOAc, dried over MgSO_4_ and concentrated under reduced pressure. The mixture was separated by column chromatography (CH_2_Cl_2_:MeOH = 10:1) to give compound **2** (36 mg, 68%): ^1^H-NMR (500 MHz, CDCl_3_) δ 7.94 (d, *J* = 8.2 Hz, 2H), 7.78 (s, 1H), 7.66 (dd, *J* = 10.9, 8.0 Hz, 2H), 7.49 (dd, *J* = 10.9, 8.0 Hz, 2H), 7.32 (d, *J* = 8.2 Hz, 2H), 6.78 (s, 1H), 6.75 (s, 1H), 6.73 (s, 1H), 5.96 (s, 1H), 5.50 (s, 1H), 5.07 (s, 1H), 5.03 (s, 1H), 4.39 (d, *J* = 12.9 Hz, 1H), 4.25 (s, 2H), 4.16 (d, *J* = 12.8 Hz, 1H), 3.88–3.79 (m, 1H), 3.60–3.48 (m, 1H), 2.93 (dt, *J* = 9.7, 6.1 Hz, 1H), 2.54 (s, 6H), 2.33 (s, 3H), 2.21–1.93 (m, 4H), 1.41 (s, 6H); ^13^C-NMR (125 MHz, CDCl_3_) δ 159.0, 143.2, 140.9, 140.3, 136.8, 135.9, 134.9, 131.4, 131.3, 131.2, 128.7, 128.1, 126.4, 121.9, 121.8, 121.4, 120.0, 115.9, 115.6, 111.9, 111.6, 69.5, 69.4, 54.8, 54.4, 29.8, 26.6, 23.6, 21.6, 21.4, 14.7 2(C); ^19^F (470 MHz, CDCl_3_) NMR δ −146.1 (m); ESI-HRMS [M + H]^+^
*m*/*z* calcd for C_42_H_46_BF_2_N_6_O_2_ 715.3743, found 715.3711.

### 3.3. Absorption and Fluorescence Spectra

Absorption spectrum was recorded at 25 °C in a 10 cm path quartz cell using a Cary 100 UV–vis spectrophotometer (Agilent, Santa Clara, CA, USA). Fluorescence spectrum was recorded at 25 °C using a Cary Eclipse fluorescence spectrophotometer (Agilent, Santa Clara, CA, USA) (cell path length: 1 cm; excitation at 492 nm). The fluorescence quantum yields (*Φ*F) were determined using a 0.1 M aqueous NaOH solution of fluorescein as a standard.

### 3.4. Sphingosine Kinase Activity Assay

The inhibition of SK activity was measured by using 100 μM of sphingosine, 10 μM of ATP, and of active SK recombinant protein (SK1: 0.5 ng/μL, and SK2: 1 ng/μL). The SK activity was measured according to the method presented in Echelon’s Sphingosine Kinase Activity Assay Kit (Echelon, Salt Lake City, UT, USA). In briefly, after mixing the compound, sphingosine and SK recombinant protein, the reaction was initiated by ATP. The reaction was terminated with a luminescent ATP-detector and read the luminescence.

### 3.5. Fluorescence Imaging

A549 cells (1 × 10^4^ cells/well) were grown in a 10 × 35 mm cell culture dish (Greiner Bio-One, Frickenhausen, Germany) for 24 h. After treatment of 10 µM BODIPY-PF-543 for 30 min cells were fixed for 10 min with 4% formaldehyde, following treatment of 10 µM Diamidine-2-phenylindole dihydrochloride (DAPI) dye for 20 min. The images were captured using ZEISS LSM710 confocal microscope. Images were analyzed using ZEN blue 2.3 lite software (Carl Zeiss Microscopy GmbH, Jena, Germany) and fluorescence intensity of the treated group was compared with non-treated cells.

### 3.6. MTT Cell Viability Assay

A549 cells viability assay was performed by 3-[4,5-dimethylthiazol-2-yl]-2,5 diphenyltetrazolium bromide (MTT) assay. Briefly, subsequent 4000 cells/well were plated in 96 well plate, after 24 h treatment of BODIY-PF-543 and PF-543, EZ-CYTOX 10 μL was added and incubated for 90 min. Absorbance was recorded using Thermo Scientific Multiskan GO at 450 nm wavelengths.

## 4. Conclusions

In the present study, we synthesized a fluoroprobed SK1 inhibitor—BODIPY-PF-543—using the abovementioned seven-step synthetic method and obtained 36 mg (overall yield, 27%) of the final Compound **2**. The maximum emission wavelength of BODIPY-PF-543 was 492 nm. The results observed for the synthesized fluoroprobed SK1 inhibitor are similar to those reported for the known BODIPY sphingolipids [25], indicating that Compound **2** is suitable as a fluoroprobe. In addition, we analyzed the SK1 inhibitory effect of BODIPY-PF-543 in comparison with that of PF-543. The IC_50_ values of BODIPY-PF-543 and PF-543 were observed to be 19.92 and 11.24 nM, respectively. Confocal microscopy after BODIPY-PF-543 treatment revealed that the compound was mainly located in the cytosol of the cells. The results demonstrate that the synthesized BODIPY-PF-543 is a valuable material for observing cell images as an SK1 inhibitor with high membrane permeability.

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
