# Peer review of "Synthesis and Biological Evaluation of BODIPY-PF-543"

_molecules, 2019, doi:10.3390/molecules24234408_

Round 1
Reviewer 1 Report
The manuscript describes the synthesis and characterization of a sphingosine-1-phosphate (S1P) inhibitor derivatized with a BODIPY dye (BODIPY-PF-543). The intracellular localization of PF-543 and the SK1 inhibitory effect has been compared with that of PF-543. Icorporation of the BODIPY moiety does not significantly affect the inhibitory effect of SK1. Confocal microscopy experiments show that BODIPY-PF-543 is mainly located in the cytosol of the cells. BODIPY-PF-543 can be used in in vitro tests (permeable to cells plus SK1 inhibitory effect).
Major corrections are required before publication:
- Extensive English revision is required.
- Abstract is too long and is highly related to the introduction section with little focus on the research developed in the manuscript.
- The term “photo probe” is often used, however, maybe the term fluoroprobe should be used instead of “photo probe”.
- It should be specified in lines 86-95 whether the choice of the BODIPY moiety as a fluorescent label or fluoroprobe is based on its emission quantum yield, or on its emission window above 500 nm, or both.
- Line 98 and below: “Compound 3” or “compound 3”? See compound 7 in lines 103-104.
- Concerning Figure 3 and lines 76-77 “PF-543 inhibits SK1 with IC50 = 2.0 nM [11].”, from Figure 3 it is difficult to understand the determination of the IC50 value reported (11.24 nM) (no minor tickmarks provided in the plot). Is the X axis label ([uM]) correct?
- Figure 4. Please, specify the meaning of DIC, CON and COMBINED+DIC ??? in the Figure caption.
- Line 141 (and 148, 157, and 168):
“3-(Azidomethyl)-5-methylphenol (4) 3-(bromomethyl)-5-methylphenyl acetate (3) (1.9 g, 0.0078 mol) was…”
3-(Azidomethyl)-5-methylphenol (4). 3-(bromomethyl)-5-methylphenyl acetate (3) (1.9 g, 0.0078 mol) was…
-Line 159: “1.92mmol) and sodium triacetoxyborohydride (272 mg, 1.28 mmol) were added thereto.”
1.92mmol) and sodium triacetoxyborohydride (STB) (272 mg, 1.28 mmol) were added thereto.
- Conclusions:
- 36 mg of final product is reported, however, the overall yield of the 7-step synthetic route should be specified as well.
- The term BODIPY-lipid is too vague. A more accurate description concerning the hydrophobic? structure of the derivatives should be used.
Author Response
Reviewer #1
The manuscript describes the synthesis and characterization of a sphingosine-1-phosphate (S1P) inhibitor derivatized with a BODIPY dye (BODIPY-PF-543). The intracellular localization of PF-543 and the SK1 inhibitory effect has been compared with that of PF-543. Icorporation of the BODIPY moiety does not significantly affect the inhibitory effect of SK1. Confocal microscopy experiments show that BODIPY-PF-543 is mainly located in the cytosol of the cells. BODIPY-PF-543 can be used in in vitro tests (permeable to cells plus SK1 inhibitory effect).
Major corrections are required before publication:
- Extensive English revision is required.
Answer: We paid careful attention to English in the manuscript. Also, the manuscript was edited by a native English speaker.
- Abstract is too long and is highly related to the introduction section with little focus on the research developed in the manuscript.
Answer: We understand the reviewer’s point. We changed the abstract in the revised manuscript.
- The term “photo probe” is often used, however, maybe the term fluoroprobe should be used instead of “photo probe”.
Answer: We changed photo probe to fluoroprobe in the revised manuscript.
- It should be specified in lines 86-95 whether the choice of the BODIPY moiety as a fluorescent label or fluoroprobe is based on its emission quantum yield, or on its emission window above 500 nm, or both.
Answer: As the reviewers pointed out, we specified the details in the revised manuscript as follows and added references.
“BODIPY is one of the excellent probes with high absorption and fluorescence in the visible range, photochemical stability, and chemical robustness. BODIPY-based probes have low toxicity and are stable at physiological pH, thus making them excellent probes for use in living cells.”
- Line 98 and below: “Compound 3” or “compound 3”? See compound 7 in lines 103-104.
Answer: We changed the compound name consistently in the revised manuscript.
- Concerning Figure 3 and lines 76-77 “PF-543 inhibits SK1 with IC50 = 2.0 nM [11].”, from Figure 3 it is difficult to understand the determination of the IC50 value reported (11.24 nM) (no minor tickmarks provided in the plot). Is the X axis label ([uM]) correct.
Answer: The IC50 value may vary slightly depending on the measurement method. For this reason, three measurement methods were used in the Pfizer (Ref: J. Med. Chem. 2017, 60, 2562−2572). The X-axis label has been corrected in the revised manuscript.
- Figure 4. Please, specify the meaning of DIC, CON and COMBINED+DIC ??? in the Figure caption.
Answer: The authors express thanks to the reviewer for the helpful comment. We corrected the figure caption according to the reviewer's point of view.
- Line 141 (and 148, 157, and 168):
“3-(Azidomethyl)-5-methylphenol (4) 3-(bromomethyl)-5-methylphenyl acetate (3) (1.9 g, 0.0078 mol) was…”
-Line 159: “1.92mmol) and sodium triacetoxyborohydride (272 mg, 1.28 mmol) were added thereto.”
1.92mmol) and sodium triacetoxyborohydride (STB) (272 mg, 1.28 mmol) were added thereto.
- Conclusions:
- 36 mg of final product is reported, however, the overall yield of the 7-step synthetic route should be specified as well.
Answer: As the reviewer suggested, we corrected it according to the reviewer's point of view.
- The term BODIPY-lipid is too vague. A more accurate description concerning the hydrophobic? structure of the derivatives should be used.
Answer: As the reviewer commented, we changed BODIPY-lipid to BODIPY-sphingolipid in the revised manuscript.
Reviewer 2 Report
The authors describe the generation of a fluorescent-labeled sphingosine kinase inhibitor and claim its applicability in vitro. Despite the very exciting prospect of using such labeled inhibitors in vitro, the authors lack to show it's functionality. Thus, the major weakness of the presented study is the lacking proof of in vitro applicability. Considering the content of introduction and abstract, the results presented in the manuscript are not sufficient to support the claimed advantage of fluorescent-tagged sphingosine kinase inhibitors.
The authors elaborate on sphingosine kinase inhibition and its role in apoptosis and thus, it's usefulness in cancer therapy. However, the authors did not test any cytotoxicity or effect on cell proliferation etc. of their fluorescence-tagged compound. Along the same lines, the authors should have tested sphingosine kinase activation in compound-treated cells, or treated cells with fluorescence-tagged compound after SK1 activation to compare with baseline. It would also be important to test an SK2 inhibitor to evaluate specificity in similar approaches as stated under point 2. It is unclear how specific SK1 activity was determined using the sphingosine kinase assay. Wouldn't this assay similarly pickup the activity of SK2? It is unclear how stable the integrated fluorescence is. Would it be suitable for live cell imaging? There are several flaws in the introduction (lines 65-80). What do the authors mean by "FTY720 type SK inhibitors, which are also S1P modulators"? It is unclear what the following sentence means: "FTY720 (Gilenya), a treatment for multiple sclerosis developed by Novartis, regulates S1P (Figure 1)". FTY720 is used in MS because of its S1Pr1-mediated immune-modulatory capacity, which is unrelated to S1P levels. "FTY720 is modified by SK2 to FTY720-phosphate (FTY720-P) and may inhibit SK1". Here it is unclear which of the two compounds inhibit SK1? "FTY720 has stronger anticancer effect in vitro than existing anticancer drugs, but it is difficult to develop into anticancer drugs based on it because it is transformed into FTY720-P by SK2". The limitation here is unclear since SK2 inhibitors are currently used in phase II clinical trials for solid cancer treatment. "Despite the physiological activities of which the mechanism is unknown, FTY720 is successfully sold in the pharmaceutical market and sells the most among the products of Novartis." FTY720 is used in MS because of its S1Pr1-mediated immune-modulatory capacity. It's physiology in this respect is clear "For this, we introduced photo probes to the PF-543 and conducted biological studies using them." The presented in vitro tests cannot be considered biological studies that test the applicability of the developed compound for cancer studies or in vitro studies at all. Overall, The introduction needs to be refined and funnelled towards a clear hypothesis or aim. Several times throughout the manuscript, the authors falsely refer to Figure 1. It is unclear why some of the figures are need figures and others are schemes. The format of the manuscript should be refined (introduction, results, discussion, conclusions, methods).Author Response
The authors describe the generation of a fluorescent-labeled sphingosine kinase inhibitor and claim its applicability in vitro. Despite the very exciting prospect of using such labeled inhibitors in vitro, the authors lack to show it's functionality. Thus, the major weakness of the presented study is the lacking proof of in vitro applicability. Considering the content of introduction and abstract, the results presented in the manuscript are not sufficient to support the claimed advantage of fluorescent-tagged sphingosine kinase inhibitors.
The authors elaborate on sphingosine kinase inhibition and its role in apoptosis and thus, it's usefulness in cancer therapy. However, the authors did not test any cytotoxicity or effect on cell proliferation etc. of their fluorescence-tagged compound.
Answer: The authors express thanks to the reviewer for the helpful comment. As the reviewer commented, we performed the cell cytotoxic effect experiment using A549 cell and added these data to the revised manuscript (Supplementary data Figure S2).
Along the same lines, the authors should have tested sphingosine kinase activation in compound-treated cells, or treated cells with fluorescence-tagged compound after SK1 activation to compare with baseline. It would also be important to test an SK2 inhibitor to evaluate specificity in similar approaches as stated under point 2.
It is unclear how specific SK1 activity was determined using the sphingosine kinase assay. Wouldn't this assay similarly pickup the activity of SK2?
Answer: We measured the SK2 activity to compare the inhibitory efficacy of the two compounds. Both compounds showed similar inhibitory activity against SK2 at the same concentration and added this data to the Supplementary data Figure S1. The SK1 activity was measured using recombinant human SK1.
It is unclear how stable the integrated fluorescence is. Would it be suitable for live cell imaging?
Answer: BODIPY is one of the excellent probes with high absorption and fluorescence in the visible range, photochemical stability, and chemical robustness. BODIPY-based probes have low toxicity and are stable at physiological pH, thus making them excellent probes for use in living cells. We added this sentence and related references to the revised manuscript.
There are several flaws in the introduction (lines 65-80).
What do the authors mean by "FTY720 type SK inhibitors, which are also S1P modulators"? It is unclear what the following sentence means: "FTY720 (Gilenya), a treatment for multiple sclerosis developed by Novartis, regulates S1P (Figure 1)". FTY720 is used in MS because of its S1Pr1-mediated immune-modulatory capacity, which is unrelated to S1P levels. "FTY720 is modified by SK2 to FTY720-phosphate (FTY720-P) and may inhibit SK1". Here it is unclear which of the two compounds inhibit SK1? "FTY720 has stronger anticancer effect in vitro than existing anticancer drugs, but it is difficult to develop into anticancer drugs based on it because it is transformed into FTY720-P by SK2". The limitation here is unclear since SK2 inhibitors are currently used in phase II clinical trials for solid cancer treatment. "Despite the physiological activities of which the mechanism is unknown, FTY720 is successfully sold in the pharmaceutical market and sells the most among the products of Novartis." FTY720 is used in MS because of its S1Pr1-mediated immune-modulatory capacity. It's physiology in this respect is clear "For this, we introduced photo probes to the PF-543 and conducted biological studies using them." The presented in vitro tests cannot be considered biological studies that test the applicability of the developed compound for cancer studies or in vitro studies at all. Overall, The introduction needs to be refined and funnelled towards a clear hypothesis or aim. Several times throughout the manuscript, the authors falsely refer to Figure 1. It is unclear why some of the figures are need figures and others are schemes. The format of the manuscript should be refined (introduction, results, discussion, conclusions, methods).Answer: We rewrote with a clearer concept in the revised version to reflect the reviewer's point of view as much as possible. In addition, the manuscript was edited by a native English speaker to improve the flow of sentences.
Round 2
Reviewer 2 Report
The authors have revised the manuscript and added data.
1. However, there are still several flaws in the introduction (lines 65-80).
A. Therefore, SK inhibitors which are structurally similar to FTY720 have been developed [7]. FTY720 (Gilenya, Figure 1), a treatment for multiple sclerosis developed by Novartis, regulates the S1P receptor and may inhibit SK1 [8].
FTY720 doesn't regulate the receptor. It is a functional agonist for S1Pr1 and antagonist for S1Pr3-5.
B. Although the in vitro anticancer effect of FTY720 is stronger than those of the existing anticancer drugs, it is difficult to develop anticancer drugs based on it because it is transformed into FTY720-P by SK2 and FTY720-P is unable to induce cell death [10]. FTY720 is still limited in its application to anticancer drugs, but has a variety of biological activities, successfully sold in the pharmaceutical market, and is one of the best- selling products manufactured by Novartis.
Not sure where the stinger anti-cancer effects of FTY720 were reported. But if that is true, a reference needs to be added. Maybe it would be advisable to be more specific than purely stating "anti-cancer" effect. The second part of this sentence is unclear.
The message of the second sentence is unclear. FTY720 is selling well because it is used in MS therapy due to its S1Pr1-mediated immune-modulatory capacity. Why is the application in cancer therapy limited? An explanation that is understandable needs to be added.
2. How exactly was Sk2 activity determined?
3. How was viability determined in A549 cells?
4. Why were compound concentrations not matched for SK2 activity and viability assays?
Author Response
Comments and Suggestions for Authors
The authors have revised the manuscript and added data.
However, there are still several flaws in the introduction (lines 65-80). Therefore, SK inhibitors which are structurally similar to FTY720 have been developed [7]. FTY720 (Gilenya, Figure 1), a treatment for multiple sclerosis developed by Novartis, regulates the S1P receptor and may inhibit SK1 [8].FTY720 doesn't regulate the receptor. It is a functional agonist for S1Pr1 and antagonist for S1Pr3-5.
Answer: The authors express thanks to the reviewer for the helpful comments. we corrected the sentence in the revised manuscript.
Although the in vitro anticancer effect of FTY720 is stronger than those of the existing anticancer drugs, it is difficult to develop anticancer drugs based on it because it is transformed into FTY720-P by SK2 and FTY720-P is unable to induce cell death [10]. FTY720 is still limited in its application to anticancer drugs, but has a variety of biological activities, successfully sold in the pharmaceutical market, and is one of the best- selling products manufactured by Novartis.
Not sure where the stinger anti-cancer effects of FTY720 were reported. But if that is true, a reference needs to be added. Maybe it would be advisable to be more specific than purely stating "anti-cancer" effect. The second part of this sentence is unclear.
The message of the second sentence is unclear. FTY720 is selling well because it is used in MS therapy due to its S1Pr1-mediated immune-modulatory capacity. Why is the application in cancer therapy limited? An explanation that is understandable needs to be added.
Answer: We thought that the reviewer's point was reasonable and thoughtfully changed many sentences in the revised version.
How exactly was Sk2 activity determined?
Answer: We have added information on how SK2 activity is measured in the method section of the revised manuscript.
How was viability determined in A549 cells?Answer: The viability method pointed out by reviewer was added in revised version
Why were compound concentrations not matched for SK2 activity and viability assays?
Answer: Both compounds were considered to have a higher selectivity for inhibition of SK1 than SK2 inhibition, so that SK 2 activity was observed by selecting a high concentration of 2 points. The cell growth inhibitory effect was thought to be due to the inhibition of SK1 activity and therefore viability assays were tested in various concentrations.